# Potential of *Jatropha curcas* L. as Biodiesel Feedstock in Malaysia: A Concise Review

**Nurul Husna Che Hamzah [1], Nozieana Khairuddin [1,\*], Bazlul Mobin Siddique [2] and Mohd Ali Hassan [3]**

1   Department of Basic Science and Engineering, Faculty of Agriculture and Food Sciences, Universiti Putra Malaysia Bintulu Sarawak Campus, Bintulu 97008, Sarawak, Malaysia; gs56633@student.upm.edu.my

2   School of Chemical Engineering and Science, Swinburne University of Technology Sarawak Campus, Kuching 93350, Sarawak, Malaysia; msiddique@swinburne.edu.my

3   Department of Bioprocess Technology, Faculty of Biotechnology and Biomolecular Sciences, Serdang 43400, Selangor, Malaysia; alihas@upm.edu.my

\*   Correspondence: nozieana@upm.edu.my; Tel.:+60-86-855-823

**Abstract:** Fluctuation in fossil fuel prices and the increasing awareness of environmental degradation have prompted the search for alternatives from renewable energy sources. Biodiesel is the most efficient alternative to fossil fuel substitution because it can be properly modified for current diesel engines. It is a vegetable oil-based fuel with similar properties to petroleum diesel. Generally, biodiesel is a non-toxic, biodegradable, and highly efficient alternative for fossil fuel substitution. In Malaysia, oil palm is considered as the most valuable commodity crop and gives a high economic return to the country. However, the ethical challenge of food or fuel makes palm oil not an ideal feedstock for biodiesel production. Therefore, attention is shifted to non-edible feedstock like *Jatropha curcas Linnaeus* (*Jatropha curcas* L.). It is an inedible oil-bearing crop that can be processed into biodiesel. It has a high-seed yield that could be continually produced for up to 50 years. Furthermore, its utilization will have zero impact on food sources since the oil is poisonous for human and animal consumption. However, Jatropha biodiesel is still in its preliminary phase compared to palm oil-based biodiesel in Malaysia due to a lack of research and development. Therefore, this paper emphasizes the potential of *Jatropha curcas* as an eco-friendly biodiesel feedstock to promote socio-economic development and meet significantly growing energy demands even though the challenges for its implementation as a national biodiesel program might be longer.

**Keywords:** non-edible; oil; biodiesel production; fuel

## 1. Introduction

The depletion of crude oil reserves coupled with the awareness of environmental issues and escalating petroleum prices have stimulated the search for alternatives to reduce overdependence on conventional fossil fuels [1,2]. Historically, researchers have substituted conventional fuels with renewable energy resources (e.g., biofuels) since the invention of diesel engines [3]. These technologies have since advanced to this day. Typically, conventional diesel and petroleum fuels release harmful gases into the atmosphere, thereby causing global warming and climate change. Furthermore, fossil fuels contribute to pollution through the emission of major greenhouse gases (GHG). In principle, a biofuel is cleaner than any fossil fuel, since it can reduce carbon dioxide ($CO_2$) emissions by 78% and carbon monoxide (CO) emissions by 50% [4].

The global human population is predicted to increase by 34% by 2050 [5]. This increase in human population has become a contributing factor to the high demand for energy consumption. Moreover,

the continuous exploitation and rapid depletion of the Earth's natural and mineral resources will significantly increase the energy demand for transportation, industrial, and other purposes. According to the International Energy Agency (IEA), it is estimated that global energy consumption will soar by 53% by the year 2030 [6]. Another study also reported that global petroleum resources will be completely depleted within 40 years [7]. This situation could result in annual oil price escalation in the near future. Therefore, renewable energy resources such as biofuels urgently need to be adopted as substitutes for conventional fuels.

Biodiesel derived from *Jatropha curcas*, which is suitably planted in tropical or subtropical countries such as Malaysia and Indonesia, could potentially reduce the use of conventional fuels. However, there are many challenges faced during the cultivation, harvesting, and processing of the crop yield. Hence, finding the root cause is important for resolving the issues and ensuring a higher quality of harvested Jatropha seeds. Consequently, Jatropha's enormous potential in the financial, agricultural, environmental, sustainable energy production, and industrial fronts could attract the attention of researchers and policymakers.

## 2. Distribution and Physicochemical Properties of Biodiesel Feedstock

Bioethanol, biodiesel, and biogas are the main biofuel components of various agricultural biomasses produced from the different biochemical routes [8]. The first generation feedstock was produced from edible oils such as corn, sugarcane, sugar-beet, and others. Unlike the first generation of biofuels, second-generation biofuels targeted non-food biomass and agricultural residues [9]. Biodiesel originated from the first and second-generation of biofuels, as it is processed from agricultural crops and residues. Typically, the sources of biodiesel differ in many regions or countries. European nations use rapeseed due to the surpluses from edible oil production. On the other hand, soybean is commonly utilized for biodiesel production in the United States, which is becoming the main biodiesel-producing country [10]. Singh, V. et al., had summarized all classification of biofuels production starting from the first until the fourth generation as shown in Figure 1.

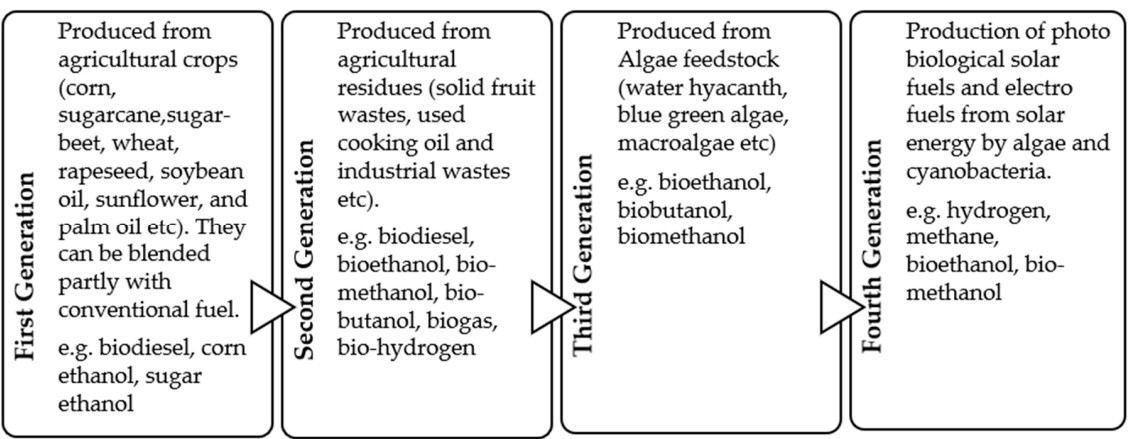

**Figure 1.** Classification of biofuels (adapted from [11]).

On the contrary, the excess palm oil and coconut oil in Malaysia, Indonesia, and Thailand could be utilized for the synthesis of biodiesel. However, the food versus fuel competition could be overcome by exploring non-edible seed oils such as *Jatropha curcas* and *Karanja* (*Pongamia* second-generation) as raw materials for biodiesel production [12]. Other than vegetable-based biodiesel, waste cooking oil (WCO) is becoming more popular for biodiesel production in Malaysia due to the low cost and the high volume of waste generation in each household. Kabir et al. reported that the average WCO generated in Malaysia per household is 2.34 kg/month [13].

Biodiesel processed from animal fats and vegetable oils is defined as fatty acid alkyl esters or fatty acid methyl esters (FAME) [14]. It is typically synthesized from the reaction of triglycerides with

alcohol and a catalyst in a process known as transesterification (Figure 2). The transesterification will produce FAME and simultaneously cause saponficiation or soap formation to occur. Common catalysts usually employed during the reaction are homogeneous alkaline catalysts—for instance, potassium hydroxide (KOH), sodium hydroxide (NaOH), potassium methoxide ($CH_3OK$), and sodium methoxide ($CH_3ONa$). However, it is necessary to control the amount of alkali catalyst, because excess alkali enhances the saponification reaction which reduces the yield of product [15]. Nevertheless, the saponification value is important as an indicator of the oil as normal triglycerides, making the oil useful in the soap and shampoo industries [16]. It is reported that the nature of the catalyst employed is crucial to converting triglycerides into biodiesel. For instance, using a homogeneous catalyst will produce glycerol or soap as a by-product, which could risk consuming or even deactivating the catalyst. As a result, the biodiesel purification process would be hampered by the loss of catalytic process [17]. As for heterogeneous catalysts, a higher quality of FAME can be generated after transesterification.

**Figure 2.** Transesterification of triglycerides with alcohol and catalyst [18].

The physicochemical properties of biodiesel depend on numerous factors—for example, the composition of the fatty acid in the raw feedstock, the chain length of the fatty acid, the saturation degree, and branching. Other factors include the production technique and operating conditions for the biodiesel synthesis [19]. The quality of biodiesel may also differ due to the impurities from unreacted feedstock glycerides, the fraction of non-fatty acids, or runaway reactions during the process of transesterification [20]. Typically, a longer fatty acid chain will enhance the synthesis of biodiesel products with a higher cetane number, which results in lower emissions of toxic NOx [21]. The composition of the fatty acid determines the level of saturation with higher compositions, resulting in a higher degree of saturation and viscosity [10].

## 3. Potential of *Jatropha curcas* as Biodiesel Feedstock

Energy crops are specifically grown for fuel and energy production. Currently, these crops only contribute to a comparably small percentage of the total biomass energy produced each year. However, this percentage is expected to increase over the next few decades. Nevertheless, energy crops compete for land earmarked for food production, environmental protection, and forestry or nature conservation. Generally, the characteristics of ideal energy crops are high yield (maximum dry matter per hectare production), low cost, low energy input, low nutrient or fertilizer requirements, pest resistance, and the composition with the least contaminants generated [22].

In Malaysia, palm oil is considered the most valuable energy crop due to its abundance and productivity. The Malaysian Palm Oil Board (MPOB) reports that the total area of oil palm planted in 2019 was 5.9 million ha [23]. Recently, the Ministry of Primary Industries and the MPOB launched the B20 Biodiesel program (20% palm methyl esters blended with 80% petroleum diesel) to offset the palm vegetable oil demand and stabilize the market price of these products [24]. However, this crop is still perceived as a major source of vegetable oil all over the world rather than as a fuel, and the demand for it as a food ingredient is increasing. Therefore, the idea of utilizing inedible food crops such as *Jatropha curcas* could help overcome the major problems faced by the first generation of biodiesel feedstock.—for example, the food vs. fuel dilemma, the issues of scaling, and the inability to grow on peripheral areas of land, etc. Its average productive life span is also longer than that of oil palm

(50 years and 30 years, respectively). Besides, its oil content is reportedly between 63.16% and 66.4%, which is higher than that of soybean (18.35%), linseed (33.33%), and palm kernel (44.6%) [25].

The tree is a drought-resistant perennial and grows well in marginal land that has little or no agricultural or industrial value due to poor soil and other undesirable characteristics. Correspondingly, unlike some other conventional edible feedstock, the planting of this crop is no threat to existing arable land land and the food chain. In 2012, it was reported that Forest Research Institute Malaysia (FRIM) has successfully planted 6000 *Jatropha curcas* plants in Terengganu, since the state has about 71,000 ha of problematic land along the coast that is left without commercialized agricultural activities [26].

Besides FRIM, other government agencies such as the Malaysian Rubber Board (MRB) estimate that approximately 50 hectares (ha) of *Jatropha curcas* were planted in Sungai Buloh, Selangor, and Kota Tinggi as of June 2012. In the early phase, about 1712 ha of land in total was earmarked for the principal cultivation of this crop in the country. Likewise, a small number of local private companies have indicated their willingness to cultivate *Jatropha curcas* on the scale of 400 ha to 1000 ha. Some stakeholders are planning to expand the cultivation to 57,601 ha in total by the year 2015. The Plantation, Industries, and Commodities Ministry (MPIC) has also initiated an experimental project on Jatropha, for which 300 ha has been allocated [4].

Biodiesel fuel has been widely adopted in most countries because it is biodegradable, non-toxic, and environmentally friendly with lower greenhouse gas (GHG) emissions. Furthermore, biodiesel adoption is considered an excellent method for reducing noise and potentially scaling down air pollutants such as carbon monoxide (CO), sulfur, polycyclic aromatic hydrocarbon (PAH), smoke, and particulate matter (PM) [27]. The most significant fuel characteristics considered for biodiesel application in diesel engines are density, viscosity, cetane number, and flash point [28].

The Cetane number is the principal indicator of fuel quality, particularly ignition and combustion in diesel engines. Typically, a high Cetane number indicates a lower ignition delay time—i.e., the time interval from the injection of the fuel to initialization of ignition in the combustion chamber. Typically, the parameter ensures a good quality fuel combustion, cold start, and engine performance, along with low white smoke formation and emissions [29]. The Cetane number of Jatropha is reported to be as high as 55, which is similar to that of diesel (Table 1). Hence, any biodiesel to be effectively substituted for diesel should retain a higher cetane number.

**Table 1.** Comparison of vegetable oil with biodiesel specification [30–33].

| Properties | Diesel | Palm Biodiesel | Jatropha Biodiesel | ASTM D6751 | EN 14214 |
|---|---|---|---|---|---|
| Cetane number | 45–55 | 52 | 57 | Min. 47 | Min. 51 |
| Flash point, °C | 50–98 | 181 | 135 | Min. 130 | Min. 120 |
| Viscosity, mm$^2$/s | 2.5–5.7 | 4.9 | 4.8 | 1.9–6.0 | 3.5–5.0 |
| Density, kg/m$^3$ | 816–840 | 879.3 | 862 | 860–900 | 860–900 |

In practice, the blend of any vegetable-based biodiesel with petroleum diesel need to comply with the two most referred biodiesel standards, namely, the American Standard Specifications for Biodiesel Fuel (B100) Blend Stock for Distillate Fuels (ASTM 6751) and European Standard for Biodiesel (EN 14214). According to the both standards, biodiesel must meet the minimum flash point—i.e., above 120 °C. The flashpoint is the temperature at which a fuel begins to burn after interaction with fire. Typically, any fuel with a high flash point could result in the deposition of carbon in the combustion compartment. Since Jatropha oil has a lower flash point compared to palm oil (162 °C and 181 °C, respectively), it has a higher potential compared to palm oil as a biodiesel.

Based on Table 1, Jatropha oil has a medium viscosity between palm oil and diesel, which is good for biodiesel utilization. Typically, most vegetable oils have higher viscosities due to their high fatty acid compositions relative to petroleum diesel. The higher viscosity indicates a better lubrication of the fuel, which reduces wear on the moving mechanical parts of the engine. Ultimately, the reduced wear prevents leakage and reduces issues related to power losses and the durability of engines. Viscosity

plays an important roles in the atomization efficiency of fuel injection inside the combustion chamber, fuel droplet size distribution, and the mixture uniformity. If viscosity is too high it may lead to pump damage, filter clogging, poor combustion, and increased emissions. A higher viscosity will also lead to greater surface tension and will influence the dissolution of a liquid jet into smaller fuel droplets, which will impose a bad effect on the spray characteristic of the fuel spray injector in a diesel engine. As a result, larger size fuel droplets are injected from an injector nozzle instead of a spray of fine droplets, leading to inadequate air–fuel mixing [34,35].

Furthermore, the price of biodiesel feedstock derived from *Jatropha curcas* is considered to be the most affordable compared to other biodiesel feedstock as shown in Table 2. Its low end price will attract consumers to be using biodiesel on the road. Eventually, it would increase the market demand for biodiesel used in the transportation vehicles and will enhance the economy from people living in rural areas. The data on the price of B100 biodiesel for different feedstock in Table 2 was reported by Lim, S. and Teong, L.K. [36].

**Table 2.** A comparison of biodiesel prices from different feedstock (adapted from [36]).

| Feedstock | Price of B100 Biodiesel (USD/Tonne) |
|---|---|
| Jatropha | 400–500 |
| Palm oil | 720–750 |
| Soybean | 800–805 |
| Rapeseed | 940–965 |

## 4. Biodiesel Processing from *Jatropha curcas*

The general processing of biodiesel from *Jatropha curcas* oil involves three major steps, namely seed drying, oil extraction, and transesterification (the processing of pure vegetable oil into biodiesel), as shown in Figure 3. There are also other minor steps that are considered significant, such as the cleaning of the seeds, dehulling, and post-harvest storage. The conventional technique for recovering oils from Jatropha seeds is through the use of a mechanical screw press machine. However, a large proportion of the oil is retained in the kernel, which requires more effective ways to extract the residual oil. The most notable extraction techniques include ultrasound-assisted systems, enzyme extraction, and the utilization of catalytic materials [37]. The catalyst materials are chemicals that enhance the process of transesterification. The extraction method is closely related to the cost of mass biodiesel production in a biorefinery plant.

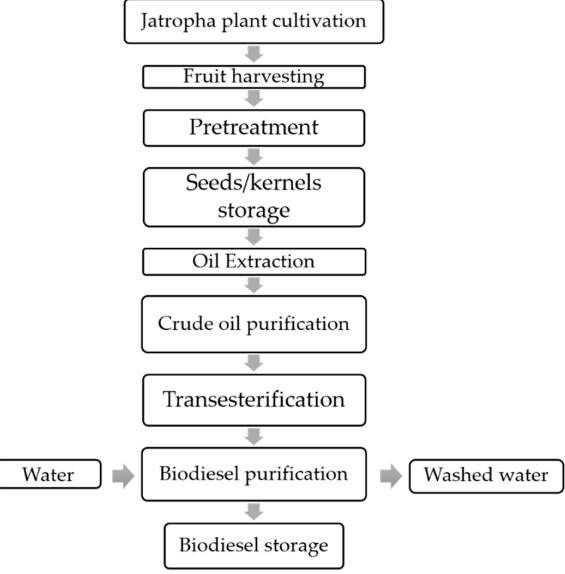

**Figure 3.** Biodiesel processing of *Jatropha curcas* [38–41].

The extracted oil subsequently undergoes a purification and transesterification process for the production of crude biodiesel. However, the crude biodiesel cannot be directly used as a transportation fuel due to limitations such as the standard requirements for biodiesel in the industry. Therefore, the crude biodiesel is usually blended with pure diesel in certain percentages before utilization in diesel engines. Before blending, the crude biodiesel is purified to remove unwanted moisture and the chemical waste produced during the transesterification process. The most popular method of purification is water washing since it is cheap and easy, although this time-consuming [42] process needs to be run several times until no more glycerol is produced.

The composition of the fatty acid significantly affects the fuel properties of the biodiesel [18]. Typically, inedible oils such as Jatropha comprise high compositions of detrimental free fatty acids (FFA) (>1% *w/w*), which reduces the biodiesel yields. Likewise, the high amount of fatty acid hampers the direct conversion of the oil into biodiesel since the high FFAs promote soap formation, which can hamper the separation of products during or after transesterification. Jatropha oil comprises nearly 14% FFA, which exceeds by far the standard limit of 1% FFA. Therefore, the pretreatment stage is required to lower the feedstock FFAs for an enhanced yield of biodiesel [30]. The typical unwanted saponification reaction that forms soap and water when NaOH catalyst is utilized is presented in Equation (1).

$$\text{R1 -COOH (FFA) + NaOH (sodium hydroxide)} \rightarrow \text{R1COONa (soap) + H}_2\text{O (water).} \qquad (1)$$

Therefore, the two-step transesterification process is an efficient method used extensively to process crude oil from *Jatropha curcas* that contains significant FFAs. Furthermore, the pretreatment or esterification process using the acid- base catalyst is performed to reduce the FFA content of *Jatropha curcas* oil. Hence, transesterification subsequently results in an optimal yield of 90% methyl ester after two hours [43]. In addition, the acid catalyst reduces the FFA content to <1% through conversion into esters by esterification. The second step involves the transesterification of the triglycerides in *J. curcas* oil into biodiesel in the presence of an alkaline catalyst. The unsaturation of fatty acids in oil is an important factor that determines the biodiesel quality. In this aspect, Malaysia, however, sits on the favourable side, as polyunsaturated fatty acids are lower in Malaysia-grown *Jatropha* oil than in the varities found in neighbouring countries [44]. Interestingly though, the Triacylglycerol (TAG) profile among these different varities of *Jatropha* oil from Malaysia and neighbouring countries did not show significant differences [45]. The kinematic viscosity of *Jatropha* oil is higher than that of regular diesel fuel, which indeed imparts a problem for all use in a diesel engine without blending. On the other side, it is much safer to handle and for storage than regular diesel fuel at higher temperatures [39]. Considering all these promising factors, its a highly promising seed oil to be taken seriously when it comes to boosting the socio-economic conditions in Malaysia.

## 5. *Jatropha curcas* Planting Challenges in Malaysia

*Jatropha curcas* is a highly promising crop for biodiesel production, although supportive innovative technologies are required for planting, harvesting, and oil extraction. Furthermore, mechanized crop operations are limited, and hence Malaysia needs to import the knowledge and machines from other countries such as India. Other notable challenges of Jatropha production are the poor seed yield, low impute crop, and pest and disease vulnerability. Although it is a promising crop for biodiesel production, the unavailability of a high-yielding cultivar is a major failure factor [46]. Besides this, high-yield fluctuation among trees and the lack of disease resistance could also hamper the commercialization of Jatropha biodiesel. In addition, it also needs appropriate nutrients and irrigation for growth and maturity, although it could flourish under limited conditions. Recent studies have reported that *Jatropha curcas* is susceptible to virus-related contaminations such as the Cucumber mosaic virus, powdery mildew, leaf spots, and soil fungous diseases. Other notable challenges are insect and rodent attacks, which result in the extensive defoliation of the plant [47,48].

In some environs, Jatropha creates complications such as weeds, which could require higher labour costs during cultivation [49]. Lastly, technologies for harvesting and post-harvesting, such as oil extraction, are still lacking. Furthermore, biodiesel is vulnerable to oxidation when exposed to air, selected storage conditions, and high levels of unsaturated fatty acids [50]. As a result, the oil content deteriorates due to inappropriate handling and storage. In addition, the main ester components of biodiesel could rapidly undergo hydrolysis to form carboxylic acids in the presence of water. Hence, both materials along with the chemical structure of biodiesel affect the swelling characteristics of the elastomer, which in turn depends on its composition and the preparation of the compound [51].

## 6. Approaches to Enhance the Jatropha Seed Oil

One of the most critical solutions is the cultivation of high oil yield *Jatropha curcas*, although such commercial varieties are lacking. The existing Jatropha breeding scheme is restricted to the traditional approach, which involves the collection of wild plant germplasm capital of Jatropha [52]. Furthermore, the review of modern applications of biotechnology for improving Jatropha is minimal [53]. In particular, research is largely absent on the expression, cloning, and annotation of biotic roles for Jatropha genes, which are responsible for its economic characteristics [46]. The main purpose of cultivation should be to advance the unit seed yield of Jatropha for commercial uses. Therefore, Jatropha cultivation techniques must require the application of numerous field practices such as planting, site planning, tree density, irrigation management, and cropping treatments.

Other notable practices involve fertilization and canopy protection, along with the control of pests and diseases. However, there is limited research that precisely and systematically validates the effect of field activity on the seed yield of *Jatropha curcas*. Selected methodological studies on planting base and management restrict the commercial cultivation of *J. curcas* [54]. Furthermore, there are limited comprehensive field or empirical reports on seed yield under different agronomic or treatment methods. For instance, data on the cultivated Jatropha tree density, the strength and interval of pruning its canopies, the insecticide impact, fertilization, and irrigation efficiency are mostly lacking in the literature [46].

## 7. Economic and Business Perspectives of Biodiesel from Jatropha Oil

Malaysia is amongst the world's premier biodiesel manufacturers. The immense profit of biodiesel in terms of rising fossil fuel values and the intention of decreasing the emission of greenhouse gasses (GHG) are factors that contribute to the development of the biodiesel market in Malaysia. Additionally, the community also is personally involved in production in order to improve income and eliminate poverty. In Malaysia, the growth of the biodiesel industry has been maintained at the top of its agenda by granting a significant amount of subsidies and farmer support programs. In fact, the government is encouraging private companies to launch more treating plants and improve biodiesel for vehicles and electricity generation. This is parallel with the post estimation of diesel vehicles, which accounts for approximately 5% of the motor vehicle population in Malaysia. The number of vehicles used is as indicated by the registered vehicles from 1996 to 2009. Based on the post estimation, diesel vehicles may possibly provide a larger share of the total in the forthcoming prior to the commencement of B5 and the campaign of the government incentives.

To date, the majority of countries have declared the standards and policies of their biodiesel. All countries have regulated their mandate or aim for biodiesel consumption success and proclaimed the exploitation of biodiesel energy fusion in their policies. As recapped in Figure 4, the national biodiesel policy of Malaysia stated on 21 March 2006 [24] objectives are as follows:

a)  The employment of environmentally friendly, sustainable, and viable sources of energy to reduce the dependency on depleting fossil fuels;
b)  The enhanced prosperity and well being of all the stakeholders in the agriculture and commodity-based industries through stable and remunerative prices;

c)　Reducing the country's dependence on depleting reserves of fossil fuels, promoting the demand for palm oil, and stabilizing its prices.

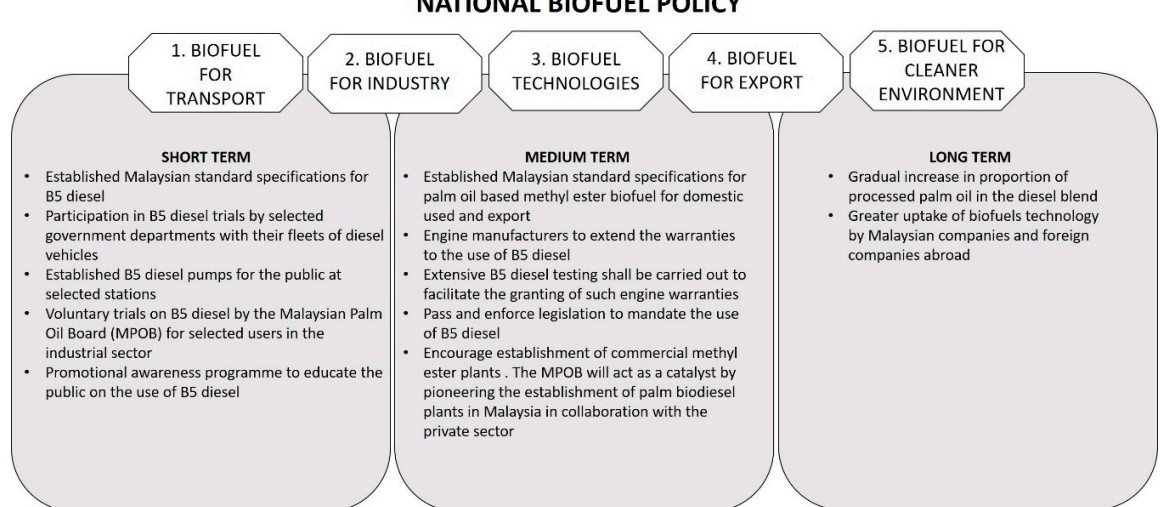

**Figure 4.** Strategic five thrusts of Malaysia's national biofuel policy and implementation (Adapted from [24]).

## 8. Conclusions

In conclusion, *Jatropha curcas* has a bright future as the next important biodiesel feedstock, considering the problems currently faced by the oil palm industry in Malaysia. It is necessary to create a higher value of its by-products in order to make Jatropha a viable biofuel in the market. Therefore, other parts of this crop, such as wood, fruit shells, seed husks and kernels, could be used to produce renewable energy [55]. The waste generated after the oil extraction process, such as the pressed cake, could also be utilized as organic fertilizer. Thus, this crop has the same characteristics as the oil palm crop, which can be used as a whole package. However, more research and development needs to be undertaken by researchers to find solutions to the existing challenges outlined in this review.

Biodiesel from *Jatropha curcas* has great potential to be implemented because it has lower carbon and emissions of GHG. It also has a lower cost compared to palm biodiesel. However, our dependency on the foreign workers in the plantation is unavoidable. The mechanization and automation specifically for maintaining the good health of *Jatropha curcas* must be improved and tested in the field beforehand. The workers must be careful while harvesting because the oil yield is dependent on the right timing of harvesting. As we know, this fruit's ripening is uneven, making harvesting a strenuous and time-consuming process. Until 2015, it has been stated that Malaysia has a total of 259,906 hectares of Jatropha crops plantation, and the current planted crops are capable of producing 4.27 tons of dry seeds every year [56,57].

As the world has been affected by global warming and the alarming threat of food security for the growing human population, much attention should be focused on non-edible oil bearing crops as biodiesel feedstock. This renewable green energy will protect the environment from the emission of harmful gases due to the combustion of fossil fuels and become an effective substitution for the depletion of the mineral resources of Earth. While many challenges await as this crop is introduced as a new biodiesel feedstock, it will never be impossible to cope with them when there is ample research and development undertaken and more expertise involved in joint ventures for the research project on *Jatropha curcas*.

**Author Contributions:** Investigation, N.H.C.H.; writing—original draft preparation, N.H.C.H.; Funding acquisition, N.K.; Supervision, N.K.; M.A.H., and B.M.S.; writing—review and editing, N.K., M.A.H. and B.M.S. All authors have read and agreed to the published version of the manuscript.

**Funding:** This research was funded by Geran Putra—Inisiatif Putra Berkumpulan grant number (9671301) to support the research and development activities in Universiti Putra Malaysia Bintulu Sarawak Campus, Malaysia.

**Acknowledgments:** The authors acknowledge the funding received from the Geran Putra—Inisiatif Putra Berkumpulan by Universiti Putra Malaysia, Malaysia.

**Conflicts of Interest:** The authors declare no conflict of interest.

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
