# Peer review of "Potential of Jatropha curcas L. as Biodiesel Feedstock in Malaysia: A Concise Review"

_processes, doi:10.3390/pr8070786_

Round 1
Reviewer 1 Report
The autors should present in the work the condition of transesterification and saponification proceses
Author Response
All comments have been responded to as attached.

Reviewer 2 Report
- What is the novel information in this paper, compared to another similar existing publications? (i.e, "Prospects of biodiesel from Jatropha in Malaysia")
- Abstract should focus less on common knowledges of biodiesel.
- Keywords are to put important words that are not mentioned in the title nor the abstract.
- The content (in particular, chapter 2, 3, 4 and 6) should focus more on current situation and prospectives in Malysia.
- Economic and business perspectives of Jatropha oil and/or biodiesel are interesting to be included (i.e., price, major challenges for Malaysian SMEs, current policy, etc)
- How expensive is the local production cost of biodiesel from Jatropha oil, compared to palm oil? Or regular diesel fuel?
- Conclusion is very weak. More highlighted facts and findings about biodiesel in Malaysia should be emphasized.
- No DOI in the reference list.
Author Response

(The authors gave the same response as above.)

Reviewer 3 Report
The article presents only basic and known information about Jatropha oil. The following important aspects were not mentioned:
- Jatropha oil is inedible. Where it will be grown? Is jatropha oil grown where palm oil is grown? This will reduce the volume of food.
- No other jatropha oil properties relevant for combustion in a diesel engine are presented: surface tension of the liquid, ability to form carbon dioxide.
- Describe correctly how viscosity, density, surface tension affect the process of preparation of the flammable mixture of jatropha oil in a diesel engine.
- Describe how the other properties of jatropha oil will affect the combustion process in a diesel engine. A diesel engine is designed to burn diesel fuel.
- Describe how other properties of jatropha oil will affect emissions of toxic components of exhaust gases - especially hydrocarbons and nitrogen oxides. What is the mechanism of this effect?
The summary of the work is too short, and it does not address the above issues. In the opinion of a reviewer, the article may be published after radical changes and additions.
Author Response

(The authors gave the same response as above.)

Round 2
Reviewer 3 Report
In its current form, the article can be printed.